# Overview of quarkonium production with ALICE at the LHC

**Hushnud Hushnud⋆ on behalf of the ALICE collaboration**

University Centre of Research and Development, Chandigarh University, Mohali, India
Department of Physics, Aligarh Muslim University, Aligarh, India

⋆ hushnud.hushnud@cern.ch

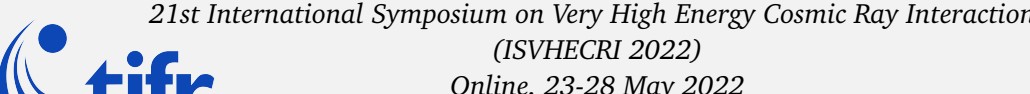

*21st International Symposium on Very High Energy Cosmic Ray Interactions
(ISVHECRI 2022)
Online, 23-28 May 2022*

## Abstract

**ALICE is a general purpose experiment designed to investigate nucleus-nucleus collisions at the Large Hadron Collider (LHC), located at CERN. The ALICE detector is optimized for the reconstruction of quarkonia through the dimuon decay channel at foward rapidity as well as the dielectron decay channel at midrapidity. In this contribution, quarkonium measurements performed by the ALICE collaboration at both midrapidity and forward rapidity for various energies and colliding systems (pp, p–Pb and Pb–Pb), will be discussed and compared to theory.**

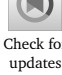

## 1 Introduction

Quarkonia, bound state of $c\bar{c}$ and $b\bar{b}$ pairs, are considered to be one of the most important probes of the deconfined hot and dense medium made of quarks and gluons, known as quark-gluon plasma (QGP), created in ultra-relativistic heavy-ion collisions [1]. In such medium, the quarkonium production yield is expected to be significantly suppressed with respect to the yield measured in pp collisions at the same centre-of-mass energy scaled by the number of binary nucleon-nucleon collisions, due to the color screening of the $q\bar{q}$ potential [1] or dissociation [2]. The temperature required for dissociating a specific quarkonium state depends on its binding energy, or equivalently on its radius. Hence, the strongly bound quarkonium states, such as $J/\psi$ and $\Upsilon(1S)$, should melt at higher temperatures compared to more loosely bound states, namely $\psi(2S)$ and $\chi_c$ for the charmonium family, and $\Upsilon(2S)$ and $\Upsilon(3S)$ for bottomonia. This is known as sequential dissociation. As a consequence, the in-medium dissociation probability of such states should provide an estimate of the medium temperature, assuming that the quarkonium dissociation is the main mechanism at play [3]. At the LHC energies, a large number of $c\bar{c}$ pairs is expected to be produced in central Pb–Pb collisions, leading to the

possibility to form charmonia via recombination of c and $\bar{c}$ quarks, either in medium [4] or at the phase boundary [5, 6]. This new additional source of charmonium production counterbalances the suppression mechanism. The regeneration mechanism has been identified as an important ingredient for the description of the observed centrality, transverse momentum ($p_T$) and rapidity ($y$) dependence of the J/$\psi$ production in Pb–Pb collisions at the LHC [7, 8]. The measurement of double ratio between $\psi$(2S) and J/$\psi$ cross sections in Pb-Pb and pp collisions at the LHC energies, is predicted to be discriminatory between recombination models.

In pp collisions, the quarkonium production can be understood as the creation of a heavy-quark pair ($q\bar{q}$) (perturbative process) followed by its hadronization into a bound state (non-perturbative process). Charmonium production measurements in pp collisions at several colliding energies are an important tool for testing various theoretical approaches involving different treatments of the non-perturbative aspects. Quarkonium measurements in pp collisions also constitute a baseline for the evaluation of the charmonium nuclear modification factor[1] in p–Pb and Pb–Pb collisions. The study of quarkonium production in p–Pb collisions can be used as a tool for a quantitative investigation of cold nuclear matter (CNM) effects such as the modifications of the parton distribution functions (PDFs) in nuclei [9, 10], $c\bar{c}$ break-up via interaction with spectator nucleons (negligible at LHC energies) [11], coherent energy loss [12] and the quarkonium dissociation due to the comoving particles ("comovers") in the final state [13]. At the LHC, the region of very small Bjorken-$x$[2] is accessible, therefore strong shadowing and coherent energy loss effects are expected.

## 2 Experimental setup and data analysis

The ALICE collaboration has studied inclusive quarkonium production in various collision systems (pp, p–Pb and Pb–Pb) down to zero transverse momentum. The details of the ALICE detector are described in [14]. Measurements are carried out at both mid and forward rapidity, in the dielectron and dimuon decay channels, respectively. Muons coming from quarkonium decays are reconstructed in the muon spectrometer, covering a pseudorapidity range $2.5 < \eta < 4$. The reconstruction of electrons from quarkonium decay is done in the central barrel within the pseudorapidity range $|\eta| < 0.9$. In the central barrel, the inner tracking system (ITS) and the time projection chamber (TPC) are used for tracking. The TPC also performs the electron identification via energy loss (dE/dx) measurements in the TPC gas. The two innermost layers of the ITS, which consist of silicon pixel detectors, provide primary and secondary vertex reconstruction which allows for the separation of the prompt and non-prompt J/$\psi$ contributions.[3] The VZERO detectors, two scintillator arrays covering the pseudorapidity intervals $2.8 \leq \eta \leq 5.1$ and $-3.7 \leq \eta \leq -1.7$, provide the minimum-bias trigger, the determination of the collision centrality and remove the beam-induced background. Two sets of zero degree calorimeters (ZDC) are used to suppress the background from electromagnetic processes in Pb–Pb collisions.

---

[1]The nuclear modification factor ($R_{AA}$) is defined as the ratio of the quarkonium yield in AA collisions with respect to the one in pp, scaled by the average number of binary nucleon-nucleon collisions.

[2]Bjorken-$x$ represents the fraction of the nucleon longitudinal momentum carried by a parton (quark or gluon) inside the nucleon.

[3]The non-prompt J/$\psi$ contribution originates from beauty-hadron decays.

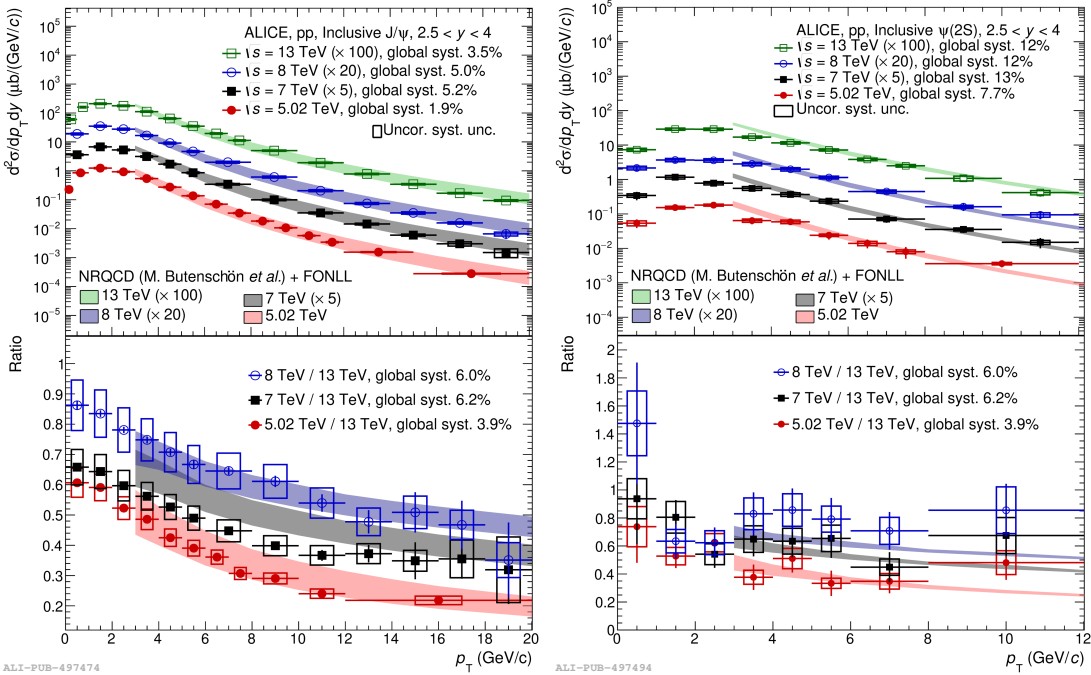

Figure 1: $p_T$ dependence of the inclusive $J/\psi$ [left] and $\psi(2S)$ [right] cross sections, measured in pp collisions at $\sqrt{s} = 5.02$, 7, 8, and 13 TeV (top panels) [15–18], and cross section ratio at $\sqrt{s} = 5.02$, 7, and 8 TeV to the 13 TeV data (bottom panels). The data are compared with NRQCD+FONLL theoretical calculations [19, 20].

## 3 Results

### 3.1 pp collisions

The $p_T$-differential cross sections of the $J/\psi$ and the $\psi(2S)$, measured in pp collisions at $\sqrt{s} = 5.02$ TeV [15] and forward rapidity ($2.5 < y < 4$), are shown in the left and right panel of Fig. 1, respectively. The results are compared with previous ALICE measurements at $\sqrt{s} = 7$, 8 and 13 TeV [16–18]. The ratios as a function of $p_T$ of the cross section measurements at 5.02, 7, and 8 TeV to the 13 TeV ones are displayed in the bottom panels of the Fig. 1. The charmonium $p_T$-differential cross sections increase with increasing collision energy. A stronger hardening of the $p_T$ spectra is observed for the $J/\psi$ (while there is no evidence for the $\psi(2S)$ in the covered $p_T$ region) at $\sqrt{s} = 13$ TeV (see bottom panels of Fig. 1).

The measured charmonium cross sections are compared with the NRQCD theoretical calculations from Butenschön *et al.* [19] for prompt $J/\psi$ and $\psi(2S)$ with FONLL calculations [20] for the non-prompt component added on top to take into account the non-prompt contributions. There is a good agreement between the model calculations and ALICE charmonium cross sections for all energies and over all $p_T$ ranges. A stronger constraint on the theoretical models can be provided by the charmonium $p_T$-differential cross section ratios among various energies because of the partial cancellation of the theoretical and experimental uncertainties.

### 3.2 p–Pb collisions

The prompt and non-prompt $J/\psi$ $R_{pPb}$, has been computed at midrapidity as a function of $p_T$ together with that of inclusive $J/\psi$ [22], as shown in Fig. 2, and compared with ATLAS results [23]. A suppression of the prompt and inclusive $J/\psi$ is observed at midrapidity and

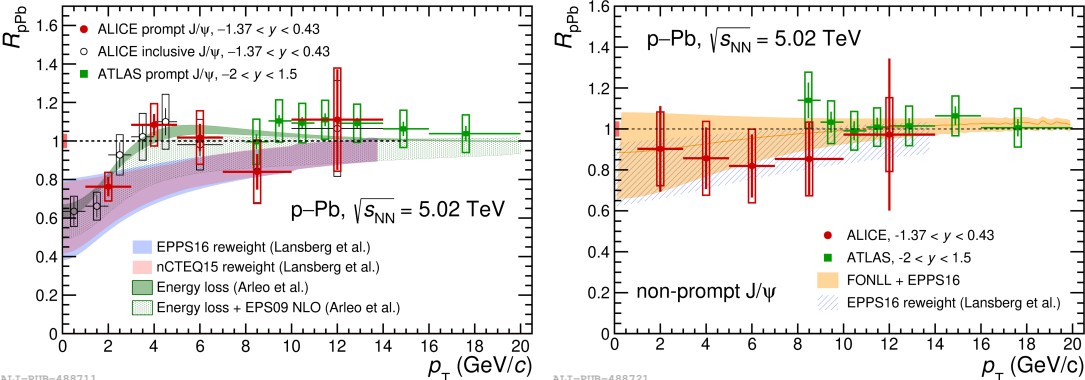

Figure 2: $R_{\mathrm{pPb}}$ of inclusive and prompt $J/\psi$ (left panel) and non-prompt $J/\psi$ (right panel) measured by the ALICE collaboration [22] as a function of $p_{\mathrm{T}}$. The results are compared with those from the ATLAS collaboration [23] and with theoretical model predictions implementing various CNM effects [12, 24–29].

low-$p_{\mathrm{T}}$ ($p_{\mathrm{T}} < 3$ GeV/$c$) as shown in the left panel of Fig. 2. The results are compared with model calculations by Lansberg *et al.* [24–27] for prompt $J/\psi$, based on the EPPS16 [28] and nCTEQ15 [29] sets of nuclear PDFs (nPDFs). Both are able to reproduce the depletion observed at low-$p_{\mathrm{T}}$. A model based on coherent energy loss [12], with or without nuclear shadowing effects included using EPS09 nPDF, provides a fairly good description of the prompt and inclusive $J/\psi$ data. In the right panel of Fig. 2, the non-prompt $J/\psi$ $R_{\mathrm{pPb}}$ results [22] are in fair agreement within the large uncertainties with the FONLL calculation employing the EPPS16 nPDFs set, which predicts a mild suppression at low-$p_{\mathrm{T}}$. Despite the large experimental uncertainties, the comparison of the prompt and non-prompt $J/\psi$ results indicate a smaller suppression without any strong $p_{\mathrm{T}}$ dependence for the non-prompt $J/\psi$. The prompt and non-prompt $J/\psi$ $R_{\mathrm{pPb}}$ measurements performed by the ALICE collaboration are found to be compatible with those of the ATLAS collaboration [23] for $p_{\mathrm{T}} \geq 8$ GeV/$c$.

## 3.3 Pb–Pb collisions

Figure 3 shows the measured nuclear modification factor $R_{\mathrm{AA}}$ of inclusive $J/\psi$ and $\psi(2S)$ as a function of centrality (left panel) and $p_{\mathrm{T}}$ (right panel). The centrality is expressed in terms of average number of nucleons participating in the collision ($\langle N_{\mathrm{part}} \rangle$). The $R_{\mathrm{AA}}$ of the $\psi(2S)$ shows no strong centrality dependence (assuming an almost constant value of about 0.4). It is significantly smaller compared to the $J/\psi$ $R_{\mathrm{AA}}$, both as a function of $p_{\mathrm{T}}$ and centrality. It also shows less suppression at low-$p_{\mathrm{T}}$ with respect to higher $p_{\mathrm{T}}$, as also observed for the $J/\psi$. This could be a first indication for $\psi(2S)$ production via recombination of $c\bar{c}$ pairs. The model calculation based on a transport approach (TAMU) [30] reproduces both the centrality and $p_{\mathrm{T}}$ dependence of the $R_{\mathrm{AA}}$ for both charmonium states.

The charmonium $R_{\mathrm{AA}}$ as a function of $p_{\mathrm{T}}$ are compared with CMS [31] measurements for $|y| < 1.6$, $6.5 < p_{\mathrm{T}} < 30$ GeV/$c$ and in the 0-100% centrality range. A strong suppression of the $\psi(2S)$ persists up to 30 GeV/$c$, as shown by the CMS data which agree very well with those of the ALICE collaboration in the common $p_{\mathrm{T}}$ range, in spite of the different rapidity coverages.

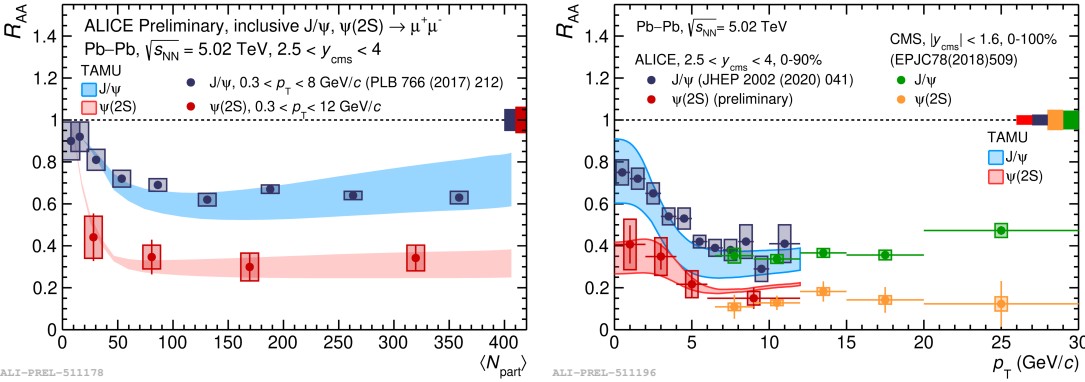

Figure 3: The $R_{\mathrm{AA}}$ of inclusive $\psi(2S)$ and J/$\psi$ as a function of the average number of participant nucleons ($\langle N_{\mathrm{part}} \rangle$) and $p_{\mathrm{T}}$, in the left and right panel, respectively. Similar results for prompt J/$\psi$ and $\psi(2S)$ from CMS [31] are also shown in the right panel. The results are compared with theoretical predictions from TAMU [30].

## 4  Conclusion

In summary, the ALICE collaboration has studied the quarkonium production in pp, p–Pb and Pb–Pb collisions at the LHC. The energy dependence of the charmonium production cross section in pp collisions has been discussed. Several model calculations describe well the quarkonium production from $\sqrt{s} = 5.02$ to 13 TeV. In p–Pb collisions at $\sqrt{s_{\mathrm{NN}}} = 5.02$ TeV, prompt J/$\psi$ shows a significant suppression for $p_{\mathrm{T}} \leq 3$ GeV/$c$, whereas less suppression is observed for the non-prompt J/$\psi$. The model predictions including various combinations of CNM effects describe well the data within uncertainties. The first accurate measurement of $\psi(2S)$ production in Pb–Pb collisions $\sqrt{s_{\mathrm{NN}}} = 5.02$ TeV has been reported at forward rapidity. The $\psi(2S)$ $R_{\mathrm{AA}}$ hints at an increase at low-$p_{\mathrm{T}}$ similar to the J/$\psi$ one and connected with charm quark recombination processes. The comparison of centrality and $p_{\mathrm{T}}$ dependent $R_{\mathrm{AA}}$ measurements of J/$\psi$ and $\psi(2S)$ with transport model, which includes recombination of charm quarks through the QGP medium, shows a fair agreement with data.

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
