# Peer review of "Overview of quarkonium production with ALICE at the LHC"

_SciPost Physics Proceedings, doi:SciPost Phys. Proc. 13, 007 (2023)_

## Round 1 · Referee Report · Anonymous (Referee 1) · 2022-12-15

Report
The author presented a very good summary of recent measurements of charmonia by ALICE in pp, pPb and PbPb collisions. This work should be accepted for publication after the following single point is more clearly explained:
In the right panel of Fig.3, the ALICE results are for inclusive (like the left panel) or prompt J/psi and psi(2S)? Especially given that the ALICE results are compared with CMS data points of prompt (I think so) J/psi and psi(2S) (at different rapidities), it is mandatory to point out all the difference between two kinds of measurements. If the ALICE measurements are for inclusive J/psi and psi(2S), then the comparison with CMS data points is not apple-to-apple, especially given that non-prompt (from bottom weak decays) contribution becomes more and more siginificant toward larger pT where CMS data points lie.
In the right panel of Fig.3, the ALICE results are for inclusive (like the left panel) or prompt J/psi and psi(2S)? Especially given that the ALICE results are compared with CMS data points of prompt (I think so) J/psi and psi(2S) (at different rapidities), it is mandatory to point out all the difference between two kinds of measurements. If the ALICE measurements are for inclusive J/psi and psi(2S), then the comparison with CMS data points is not apple-to-apple, especially given that non-prompt (from bottom weak decays) contribution becomes more and more siginificant toward larger pT where CMS data points lie.

---

## Editorial Decision

published